# Integrated Reporting Implementation and Core Activities Disclosure in UK Higher Education Institutions

Mahalaxmi Adhikariparajuli [1] , Abeer Hassan [2,*] and Mary Fletcher [2]

1 School of Business, Law and Social Sciences, Abertay University, Dundee DD1 1HG, UK; M520833@uad.ac.uk
2 School of Business & Creative Industries, University of the West of Scotland, Paisley PA1 2BE, UK; Mary.Fletcher@uws.ac.uk
* Correspondence: abeer.hassan@uws.ac.uk

**Abstract:** Through integrated reporting (IR) and integrated thinking (IT), higher education institutions (HEIs) can engage future students, staff and other key stakeholders. This paper examines the impact of IR framework implementation on core activities disclosure within the UK HEIs. In particular, the authors explore the influence of the integrated thinking approach, intended to enhance the extent of the IR content elements, teaching and learning, internationalisation and research activities disclosure. The study is based on the annual reports of 123 UK HEIs over 3 years—2015–16, 2016–17 and 2017–18. Consistent with the predictions of legitimacy theory and the integrated thinking approach, the results show the extent that integrated reporting content elements and HEIs core activities disclosure practices have increased over the study period. The results also indicate that adoption of the IR framework, league table ranking position, key performance indicator reporting, size, research quality and graduate prospects all have significant positive influences on HEIs core activities disclosure. Based on the findings, the recommendations are that UK HEI governing and other regularity bodies, such as British Universities Finance Director Groups, Leadership Foundation in Higher Education and the Higher Education Funding Council, should consider development of voluntary integrated reporting guidelines and a core activities disclosure framework.

**Keywords:** content analysis; disclosures; integrated reporting; integrated thinking; higher education institutions; legitimacy theory; value creation



## 1. Introduction

Higher education institutions (HEIs) can influence a large proportion of future leaders and make a huge impact on short, medium and long-term value creation (Adams 2018). However, HEIs are very poor at communicating their contribution and value-added activities to society, environment and governance (BUFDG 2016; Hassan et al. 2019). IR (integrated reporting) was developed as a means of improving communication between organizations, such as HEIs and their stakeholders. One stream of prior literature on IR and core activities disclosure in HEIs is very limited and carried out in single institutions. For example, IR in HEIs (Veltri and Silvestri 2015; Brusca et al. 2018; Hassan et al. 2019). The other stream of literature on IR and HEIs focuses on HEI core activities disclosure, including teaching and learning, internationalisation and research (T&L, INT and RSH) (Gordon et al. 2002; Coy and Dixon 2004; Siboni et al. 2013; Creaton and Heard-Lauréote 2019).

The main aim of this paper is to investigate whether UK HEIs are increasing their disclosure of both IR content elements and HEI core activities (T&L, INT and RSH). In this context, the main motivation of this research is to critically examine the combination of IR with HEI core activities that enables HIEs to provide more disclosure on their contribution to society, environment and governance. The current research seeks to explain the evolution of the content of corporate reporting disclosure resulting from an integrated thinking (IT) approach through IR, which has been advocated by prior scholars (Solomon and

Maroun 2012; De Villiers et al. 2014; Adams 2017; Cortesi and Vena 2019; Hassan et al. 2019). We suggest that integrated thinking is an internal process that organizations can follow to increase the level of disclosure on integrated reporting and core activities as a communication tool with stakeholders.

This research provides numerous contributions to IR practice and HEI core activities disclosure. The first contribution of this research is to investigate the IR implication of HEI core activities disclosure as suggested by several scholars (Adams 2018; Rinaldi et al. 2018; Hassan et al. 2019). Second, the researchers investigation extends the research on the influence of integrated thinking in the content of external reporting disclosure practices (SAICA 2015; Feng et al. 2017; Cortesi and Vena 2019). Third, this study links legitimacy theory with HEI voluntary core activities disclosure and investigates the motivation to adopt different reporting guidelines to increase accountability in HEIs (Maingot and Zeghal 2008; Higgins et al. 2014; Ntim et al. 2017).

Fourth, to the best of the authors' knowledge, this is the first study which combines all IR content elements, i.e., teaching and learning, internationalisation and research (T&L, INT and RSH) activities as HEI core activities disclosure practices as suggested by (Siboni et al. 2013; Adams 2018; Hassan et al. 2019). Finally, at this early stage of the IR implication journey in the public sector, existing research is largely exploratory in nature (Katsikas et al. 2017; Montecalvo et al. 2018) with one conceptual paper (Cohen and Karatzimas 2015). Therefore, little is known about the determinants of IR practice, in particular, research on the organisational sector-specific characteristics remains lacking in empirical studies (Vaz et al. 2016; Ghani et al. 2018). A significant contribution of this research is, therefore, to provide an initial empirical account of the relationship between HEI sector-specific characteristics, IR content elements and core activities disclosure.

Content analysis was employed to analyse 123 UK HEI annual reports published between 2016 and 2018. This paper will proceed as follows: the next section will explain the link between integrated reporting and integrated thinking. This is followed by a discussion on legitimacy theory in higher education. The research method is discussed in the following section, and empirical findings derived from comparative analysis of HEI annual reports. The results are discussed and in the final section, the paper draws conclusions and provides recommendations for future research.

## 2. Integrated Reporting and Integrated Thinking

Integrated reporting (IR) was developed to provide a combined disclosure of financial and non-financial information. This is achieved by the publication of a single report from the perspective of stakeholders (King Report IV 2016). The International Integrated Reporting Council (IIRC 2013) asserts that IR supports concise and comprehensive communication through the periodic integrated report about value creation over the time.

Integrated Thinking (IT), in the other hand, can be defined as the connection and interrelation between different internal functional units which influence external communication practices (King Report IV 2016). The corporation, it is argued, should develop integrated thinking within the organisation before adopting IR (Katsikas et al. 2017) where IT can drive value creation by integrating the organisational strategy, governance, performance and prospects towards the external environment. Thus, IT enables corporations to describe how they create value in a clear and meaningful manner (SAICA 2015). Furthermore, IT has been related to organisational change whereby all individual units within the organisation contribute to long-term value creation (IFAC 2017). Al-Htaybat and Alberti-Alhtaybat (2018) describe an 'Integrated Thinker' organisation as one that should deal with uncertainty and disruption based on both an individual and organisational disposition within its own field. IT and IR are strongly linked and IR is an effective mechanism of enhancing accountability, therefore all members of the organisation need to embrace the IT approach (Rinaldi et al. 2018; Higgins et al. 2019). Through an IT approach, the organisation can switch to a forward-thinking report format with clear future growth prospects (De Villiers et al. 2014).

However, very few prior scholars have investigated IT within HEIs. Among the prior contributions, Veltri and Silvestri (2015) found that implementing IT as an internal cultural and organisational mechanism, HEIs can achieve competitive advantage and consequent reputational benefit via sectorial differentiation in the South African context. In the Spanish context, researchers found that HEIs do not embed IT within the organisation and IR can be considered as a further step in the sustainability journey, which may be developed in near future but there is a long way to go to achieve this end (Brusca et al. 2018). In the UK, Adams (2018) suggests that universities have the biggest impact of any sector via sustainability reporting and are the largest beneficiaries of IT and IR. Therefore, HEIs can embed the IT approach by connecting internal management, stakeholder engagement and external reporting. Hassan et al. (2019) found that UK HEI voluntary disclosure is positively influenced by IT implication practice as an internal mechanism.

Based on the above discussion, to produce effective IR, HEIs should follow an integrated thinking approach. Our investigation of the disclosure on IR content elements and HEI core activities reveals their connectivity and interdependencies as a reflection of integrated thinking, leading to the provision of increased disclosure. This is because disclosure on the content elements (such as external environment, governance, risk and opportunities, performance, outlook, etc.) and HEI core activities brings together information from a wide range of different departments in the organization. See Figure 1 for more details.

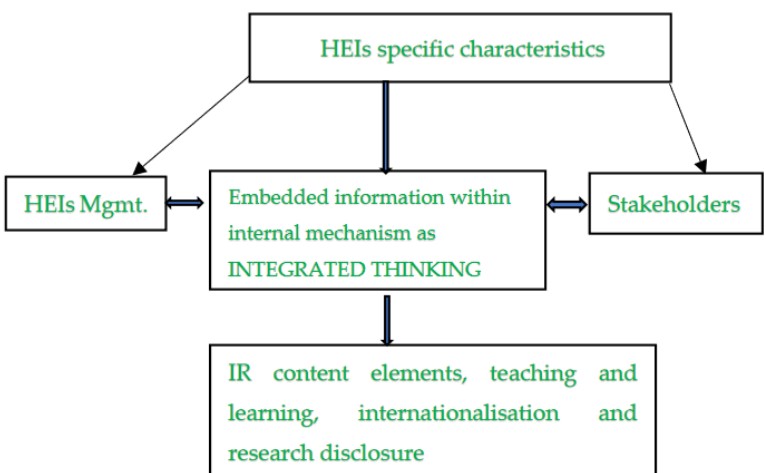

**Figure 1.** Integrated thinking and HEIs core activities disclosure.

## 3. Theoretical Framework and Literature Review

### 3.1. Theoretical Framework

In social and environmental voluntary disclosure, legitimacy theory, institutional theory, stakeholder theory and signalling theory are widely employed (Tilling 2004; Adams et al. 2016; Ntim et al. 2017; Hassan et al. 2019). Legitimacy theory was derived from political economy theory and applied to explain organisational decisions about voluntary IR (Beck et al. 2017; Haji and Anifowose 2017). Disclosure on organisational activities, it is argued, is used by organisations to enhance corporate transparency and legitimise their firms (Fernando and Lawrence 2014). The concept of 'legitimacy' is directly related to the concept of a 'social contract'. Organisations depend on an unwritten contract with their stakeholders and must be seen as legitimate in order to survive. As society changes, organisations can experience difficulties in attracting capital, employees and customers, but effective managers are able to anticipate changing community expectations and preferences (Deegan 2006). Whenever a manager considers that a supply is vital for organisational survival, then the manager can change strategies such as targeting disclosure, controlling disclosure or collaborating with other parties (Deegan 2002).

According to legitimacy theory, organisations adopt voluntary social and environmental disclosure and IR to survive within their sectors and fulfil stakeholder expectations

by bridging the legitimacy gap (Higgins et al. 2014; Ntim et al. 2017). In the HEI sector, scholars have employed legitimacy theory to investigate voluntary disclosure practices. For example, Chatelain-Ponroy and Morin-Delerm (2016) argue that some universities that adopted sustainable development and voluntary disclosure did so in order to obtain social approval. Taking a similar view, Ntim et al. (2017) use legitimacy theory as a basis for their prediction that HEIs will increase their voluntary disclosure for the positive consequences of institutional reputation, image and public goodwill. Therefore, in the HEI context, legitimacy theory is consistent with the asymmetry of information disclosure in favour of HEIs if they implement IT to comply to social norms and values and disclose all value creation activities done for society (Adhikariparajuli et al. 2021); this supports enhancement of their accountability (Ntim et al. 2017; Rinaldi et al. 2018).

Based on the above discussion, it is argued that legitimacy theory will be relevant because HEIs adopt the IR framework, implement integrated thinking and disclose more information about teaching learning, internationalisation and research activities to strengthen their social legitimacy.

*3.2. Literature Review*

3.2.1. Integrated Reporting Empirical Studies in Profit-Oriented Organisations

Many prior IR empirical studies have been carried out based on the IIRC framework in order to explore the practical issues of IR and its internal implications (Solomon and Maroun 2012; Pavlopoulos et al. 2017; Stubbs and Higgins 2014; Mio et al. 2016). A further group of studies link IR with CSR and SR disclosure (Stacchezzini et al. 2019; Bernardi and Stark 2018; Petcharat and Zaman 2019). The findings of these studies indicate that CSR and SR are embedded with IR and are highly influenced by the IIRC framework whilst IR implementation issues are highly experiential and call for more empirical studies (De Villiers et al. 2014; Rinaldi et al. 2018). Much of the above research has been based in developed countries, as due to scarce resources, stakeholder demands, the regulatory requirements and lack of awareness, IR implementation in developing countries is very slow (Gunarathne and Senaratne 2017; Bananuka et al. 2019; Petcharat and Zaman 2019).

Although IR implementation is still in an early stage, some of the scholars have explored IR preparation and implications via semi-structured interviews (Feng et al. 2017; Maroun 2018). In this context, some research has focused on IR auditing, assurance and materiality (Haji and Anifowose 2016; Maroun 2018; Cerbone and Maroun 2019) and the results reveal that there is still a huge gap in IR preparation, assurance and materiality disclosure.

3.2.2. Integrated Reporting Empirical Studies in the Public Sector and Higher Education Institutions

IR Empirical Studies in the Public Sector

In comparison with profit-oriented organisations, few prior studies have been carried out in the public sector and very little in the HEI sector. In relation to the public sector, Guthrie et al. (2017) explore the link between IR and IT as the internal mechanism of change in the Italian public sector. Montecalvo et al. (2018) examine the influence of IR on the sustainability practices in the enterprises owned by the state governments in New Zealand. As a case study in a university hospital, Cavicchi et al. (2019) investigate IR mechanisms that affected the potential development of IR practice in the Italian health care sector and Tirado-Valencia et al. (2019) explore the implication of IT in state-owned entities and conclude that IT is not fully implemented during integrated report preparation. External factors influenced value creation frequently and the connectivity of related information sources meet stakeholder demands was not addressed.

IR Empirical Studies in HEIs

In relation to the HEI sector, Veltri and Silvestri (2015) conducted a pioneering study into The Free State University (TFSU) IR in South Africa for academic year of 2012 and the

results reveal that the university corporate report does not follow the IIRC guidelines and none of the future outlook aspects are disclosed. IR and SR in the HEIs in the voluntary reporting context of an innovative Spanish university are explored by Brusca et al. (2018). The results indicate that the report mainly focused on sustainability and social value, rather than connecting all capitals from the IR framework and IT was not embedded within the organisational report. Hassan et al. (2019) examined the HEI sector-specific characteristics and IR content elements disclosure in UK HEIs and the results revealed that HEI IR content elements disclosure is significantly and positively associated with the length of time of HEI establishment, IR framework adoption and HEI size. Although there is a lack of research on IR in the HEI sector, there is a considerable body of research on individual HEI activities which is discussed in the following section.

### 3.3. Teaching and Learning, Internationalisation and Research Disclosure

In the context of Teaching and Learning (T&L), Internationalisation (INT) and Research (RSH) activities, a large body of literature has investigated T&L, INT and RSH in HEIs based in different countries. In Australia (Gao 2018); in New Zealand (Coy and Dixon 2004; Elkin et al. 2005); in Canada (James-MacEachern and Yun 2017); in Italy (Siboni et al. 2013; Bratti and Verzillo 2019); in Nordic universities (Kristensen and Karlsen 2018), and in the UK (Lomas 2006; Sengupta and Ray 2017; Johnston and Reeves 2019). However, much less research has been based on UK HEI annual reports (Ntim et al. 2017; Hassan et al. 2019; Adhikariparajuli et al. 2021). Our study therefore focuses on UK HEI annual reports as a key source of disclosure.

Prior research on HEI disclosure in the UK and elsewhere has employed a range of approaches. Some researchers carried out qualitative research for T&L, INT and RSH disclosure (Lomas 2006; Ibrahim 2014; Kristensen and Karlsen 2018; Brennan et al. 2019). Others have used quantitative approaches (Ntim et al. 2017; Wu and Jessop 2018; Malfroy and Willis 2018; Johnston and Reeves 2019). In this study, we are trying to include HEI core activities T&L, INT and RSH to be the part of IR content elements.

## 4. Development of Hypotheses

### 4.1. Russell Group Membership of Higher Education Institutions

The Russell Group was formed in 1994 as a self-selecting group of UK universities seeking to differentiate themselves (Hewitt-Dundas 2012). Boliver (2013) argued that Russell Group Membership (RGM) universities tend to score highly for research output and successfully promote themselves as representing the UK leading universities. Moreover, Pursglove and Simpson (2007) summarise the debate by concluding that RGM universities appear to be more cohesive, have similar attributes, objectives and strategies, however, the academic effectiveness of post-1992 UK universities was considerably greater in comparison with RGM; the academic staff of post-1992 were more efficient than their RGM counterparts in converting their salary into disposable revenue.

Empirically, the evidence is mixed. Whilst the positive influence of RGM on voluntary disclosure was supported by Hewitt-Dundas (2012) who investigate research intensity and knowledge transfer activity in UK universities, the conclusion is that RGM universities are larger in terms of income, employees, income staff ratio and higher research incentives in comparison with other UK universities. Likewise, Sengupta and Ray (2017) engaged in a comparison between RGM and the rest of the UK HEIs with indicators of year of foundation; average staff size; research output; quantity of research-active departments; and income from knowledge transfer. They found all indicators have higher scores by RGM universities in comparison with the rest of the UK universities.

Based on the above discussion, UK RGM universities appear to perform very well in T&L, including RSH with average performance in INT. As the highest performers in the RSH sector, (obtaining higher research funding and higher endowment income) it is anticipated that RGM universities will produce clear and concise reports with IR

content elements disclosure, including T&L, INT and RSH disclosure. This led to the first hypothesis in this study:

**Hypothesis 1 (H1).** *There is a positive relationship between Russell Group member universities and the level of integrated reporting content elements, teaching and learning, internationalisation and research activities disclosure.*

### 4.2. Integrated Reporting Framework Adoption

Earlier studies related to IR framework adoption revealed contradictory results. Where Melloni et al. (2017) provide evidence that, in practice, corporations are struggling to produce a concise, balanced and complete integrated reports, Doni et al. (2019) found that the IR framework was not suitable for all organisations and they need to re-conceptualise their capitals, potentially trade-off between them and disclose net contribution to stakeholders. In addition, Cortesi and Vena (2019) suggest that entities featuring the factors that would usually motivate them towards IR adoption were not always hugely beneficial and Gunarathne and Senaratne (2017) found that IR was a transition from SR rather than a transformation and concluded that corporations need more guidance in the process to achieve IT in practice. However, other studies found that corporate transparency and future perspective reporting were increased after IR adoption (Haji and Anifowose 2017; Pavlopoulos et al. 2017; Kilic and Kuzey 2018).

In the HEI context, Veltri and Silvestri (2015) found that the State Free University's integrated reports in South Africa do not reflect the meaning and intentions of the fundamental concepts, guiding principles and content elements of IIRC framework. Brusca et al. (2018) find that HEIs do not implement IT when preparing IR in the Spanish context. In the UK, BUFDG (2016) provides evidence that universities were beginning to prepare higher-quality integrated reports though more practice was necessary in the critical analysis stage and more creativity to draw out the narrative from the figures and tell their stories. More interestingly, Hassan et al. (2019) revealed results that suggested HEI IR content elements disclosure was significantly and positively influenced by the length of HEI establishment, and by IR framework adoption and size. Therefore, this research expects that the IR framework adoption will be positively associated with IR content elements and HEI core activities disclosure. Hence the second hypothesis is formulated.

**Hypothesis 2 (H2).** *There is a positive relationship between integrated reporting framework adopted universities and the level of integrated reporting content elements, teaching and learning, internationalisation and research activities disclosure.*

### 4.3. League Table Ranking Position

League Table Ranking (LTR) performance position provides a means of evaluating universities, raises public accountability and provides guidelines to related stakeholders (Gibbons et al. 2015). The complete university guide league table (2021) group their indicators as follows: entry standards; student satisfaction; research quality; and graduate prospects. Gibbons et al. (2015) provide evidence from the National Student Survey (NSS) suggesting that LTR has a statistically significant impact on applications. The influence of LTR primarily operates through high-ability candidates demonstrating more responsiveness to rankings with changes in rank having the greatest impact on student decisions. Robinson-Garcia et al. (2014) suggests that universities with higher LTR rankings tend to offer a more complete picture of RSH performance, are supportive of policymakers and represent a snapshot of the universities and their scientific strengths. As such, Christie (2017) provides evidence that the significance of the league table for the publisher is in establishing their trustworthy status, employment measurement and to compare with other stakeholders. Allen (2019) found that, in the Chinese context, students preparing to study aboard were more curious about university ranking, had more educated parents and were also very keen to understand university ranking, demonstrating that families with higher

degrees and higher capital paid more attention to these elite status symbols. On the basis of this research, it is likely that HEIs disclose information about their indicators based on LTR and this leads to the third hypothesis:

**Hypothesis 3 (H3).** *There is a positive relationship between the league table ranking position and the level of integrated reporting content elements, teaching and learning, internationalisation and research activities disclosure.*

*4.4. Teaching Excellence Framework Performance*

Within HEIs there are bona fide interrelated concerns regarding governmental demands for increased accountability, student fees and public debt, student satisfaction, value for money and the perceived imbalance of teaching quality and research output. These concerns gave rise to the Teaching Excellence Framework (TEF), which was published as the most comprehensive form of assessment (Hayes 2017; Barkas et al. 2019). Hayes (2017) suggests that TEF metrics show how universities work towards greater equivalence of international students, which motivates them to choose the better-rated university. The nature of the metrics employed in national evaluations of teaching quality can affect the status of international students; however, Barkas et al. (2019) argued that TEF appears to be positive though its implementation appears to be conceptually flawed, demand more layers of bureaucracy in higher education and requires more empirical study related to the student experience. In addition, Ashwin (2017) concluded that TEF has the potential to provide valid information to potential students about the quality of different university courses, their performance measurement and quality of teaching. This research argues that IR content elements and core activities disclosure may be affected by TEF performance ranking and this led to the fourth hypothesis.

**Hypothesis 4 (H4).** *There is a positive relationship between the teaching excellence framework ranking position and the level of integrated reporting content elements, teaching and learning, internationalisation and research activities disclosure.*

*4.5. Key Performance Indicator Reporting*

Tee (2016) finds that Key Performance Indicators (KPIs) have been used in HEIs in assessing the performance of the whole university. This includes a better understanding of practice or performance, improved collaboration, networking and partnership, including accountabilities of universities towards stakeholders. Similarly, Chen et al. (2009) conducted comparative research between public and private universities as a case study in Taiwan. They suggested that universities use KPIs as a form of self-appraisal to measure operating performance and they were strictly designed as clear and complete to check the performance of each university. In contrast, Kairuz et al. (2016) highlighted that universities expect academic professionals to use appropriate, valid and reliable research measures. With this ethical approach of scholarly activities as performance management, they suggest that it is not necessary to disclose KPIs. Further, they claimed that KPIs and performance were detrimental to higher education.

An exploration by Cheruiyot and Maru (2013) revealed that universities who report KPIs explained their teaching process, services and infrastructure more clearly. They find a positive and significant relationship between service quality and university performance. However, reliability, assurance, tangibility, empathy and responsiveness have negative though insignificant relationships with performance.

In contrast, Lewis et al. (2007) provided evidence that, through Key Performance Indicators Reporting (KPIR), universities can disclose measurable goals and outcomes which are useful for policymakers and decision-makers, especially parents and students. KPIR can be employed for public accountability, internal institutional assessment and external-performance based funding requirements. From the prior literature, the researchers con-

cluded that KPI disclosure leads to more transparency in terms of performance, strategy and operational goals. This leads to the following hypothesis:

**Hypothesis 5 (H5).** *There is a positive relationship between key performance indicator reporting and the level of integrated reporting content elements, teaching, and learning, internationalisation and research activities disclosure.*

### 4.6. University Governing Board Size

Empirically, the evidence is mixed related to Governing Board Size (BSIZE) and voluntary disclosure. Whilst the positive influence of a large BSIZE on voluntary disclosure in the context of listed companies was indicated (Baldini and Liberatore 2016; Pavlopoulos et al. 2017), other research shows very different results. Ntim et al. (2017) find that there was no significant relationship between university BSIZE and voluntary disclosure. Hassan et al. (2019) find no significant relationship between BSIZE and IR content elements disclosure in UK HEI annual reports. Although the research findings are mixed the researchers suggest that university BSIZE has a potential influence on voluntary disclosure and the sixth hypothesis was formulated as follows.

**Hypothesis 6 (H6).** *There is a positive relationship between university governing board size and the level of disclosure of integrated reporting content elements, teaching and learning, internationalisation and research disclosure.*

### 5. Methodology

Sample Selection

The sample was based on the entire population of 130 UK HEIs listed in The Complete University Guide (2021). However, 7 universities did not publish the required information. Therefore, the research sample was limited to 123 UK HEIs that comprised the four geographical regions (England, Scotland, Wales and Northern Ireland) for the academic years 2015–2016, 2016–2017 and 2017–2018. The rationale behind choosing the year 2016 was that this was the year when BUFDG launched the first phase of their case study on UK HEIs about IR adoption. At the results stage of the BUDFG study, all 10 HEIs committed to the IR process. This identfied 2016 as the essential time period to investigate IR in UK HEIs. Moreover, TEF and Student Outcomes Framework (SOF) were published by the UK government in August 2016 and BRIXT results were published in June 2016, which influenced the HEI core activities disclosure to attract international students and professionals.

Overall, the sample was larger than prior empirical studies in HEI sector within the UK and elsewhere (Gordon et al. 2002; Brusca et al. 2018). This study collects various types of secondary data based on the IR framework published by the IIRC (2013) and HEI core activities indices from Coy and Dixon (2004) and Elkin et al. (2005). For the control variables, financial variables were primarily downloaded from selected HEI annual reports, websites and non-financial variables were collected from the Complete University Guide League Table.

### 6. Research Variables

*Integrated Reporting and Higher Education Institutions Core Activities Disclosure Checklist*

Content analysis can be used as the measurement of comparative position trends in reporting (Low et al. 2015). In addition, content analysis is essential in research as a data obtaining process by observation and analysis of the content or message of written text. The result is often a quantification of qualitative data (Krippendorff 2018). Consistent with prior literature, content analysis is a highly implemented research methodology for corporate reporting disclosure within the HEIs sector (see Gordon et al. 2002; Coy and Dixon 2004; Elkin et al. 2005; Siboni et al. 2013; Ntim et al. 2017; Hassan et al. 2019). The current study adheres to this practice to analyse UK HEI voluntary disclosure.

To construct the disclosure checklist, different sources were considered. Firstly, consistent with recent studies (BUFDG 2016; Pavlopoulos et al. 2017; Adhikariparajuli et al. 2021), this study focused on the IR content elements derived from IIRC framework (2013). Secondly, to construct T&L, INT and RSH as HEI core activities' disclosure items, the researchers followed the Public Accountability Index (PAI) introduced by Coy and Dixon (2004) and the INT index was further developed by consideration of Elkin et al. (2005). The final checklist, as shown in Appendix A, contains 11 main categories and 77 specific items that can potentially appear in HEI annual reports.

## 7. Scoring the Corporate Reports

To analyse voluntary disclosure in the HEIs sector, various scholars employ a weighted scoring approach (Maingot and Zeghal 2008; Low et al. 2015; Hassan et al. 2019; Adhikariparajuli et al. 2021). The weighted scoring approach goes beyond 'what it says' and explains 'how it says it', thus, an index assigns a weight to each item reflecting the quality of each type of related information (Haji and Anifowose 2016; Hassan et al. 2019). In contrast, the unweighted index gives each item the same score (Oliveira et al. 2006). Building on the prior literature, the current study employed the detailed scoring approach to measuring the quality of IR content elements and HEI core activities disclosure (Haji and Anifowose 2016). The researchers designed the scoring scheme of 0–3 as follows: if IR content elements and HEI core activities (T&L, INT and RSH) were not disclosed = 0. Just disclosed though not to linked with strategy, governance, performance and prospects = 1; disclosed and linked with strategy, governance, performance and prospects = 2; and disclosed and linked with strategy, governance, performance and prospects with a comparison of the historic, present and future position = 3. Under this approach, the possible highest disclosure will be 231 (i.e., $3 \times 77 = 231$). To reduce the subjectivity of coding, eight HEIs were selected as pilot study, two coders carried out the coding process independently. As a result, changes and adaptations were considered through the discussion and one researcher carried out the rest of the coding process.

## 8. Variables Measurement

### 8.1. Dependent Variable

The total disclosure score of IR content elements and HEI core activities (T&L, INT and RSH) was the dependent variable.

### 8.2. Independent Variables

HEI specific characteristics were the main independent variables for this research. The researchers collected data on:

RGM (Hemsley-Brown 2015; Boliver 2016);
IR Adoption (Solomon and Maroun 2012; Pavlopoulos et al. 2017; Hassan et al. 2019);
LTR (Gibbons et al. 2015; Christie 2017);
KPIR (Chen et al. 2009; Albats et al. 2018);
TEF Ranking Position (Wild and Berger 2016; Barkas et al. 2019);
University Governing BSIZE (Ntim et al. 2017; Pavlopoulos et al. 2017; Hassan et al. 2019).

### 8.3. Control Variables

Financial control variables

Size (Lee and Riffe 2019; Maingot and Zeghal 2008; Adhikariparajuli et al. 2021);
Annual Council Funding (IFUND) (Brusca et al. 2018; Ntim et al. 2017; Hassan et al. 2019);
Liquidity (LIQ) (Ntim et al. 2017; Hassan et al. 2019; Adhikariparajuli et al. 2021);
HEIs age (LNAGE) (Banks et al. 1997; Ntim et al. 2017).

### 8.4. Non-Financial Control Variables

All non-financial control variables employed for this study were based on the Complete University Guide as follows:

University entry standard (ES) (Ayoubi and Massoud 2012; Boliver 2015);
Research quality (RQ) (Rozman and Marhl 2008);
Student satisfaction (SS) (Douglas et al. 2006);
Graduate prospects (GP) (Finch et al. 2013)

The researchers decided to employ both financial and non-financial control variables as organisational voluntary disclosure can be affected by both (Veltri and Teresa Nardo 2013a; Nardo and Veltri 2014; Brusca et al. 2018; Pavlopoulos et al. 2017). Table 1 demonstrates the research variables used in H1–H6 or the 3 years from 2016 to 2018.

**Table 1.** Summary of variables and measurement.

| Variables | Acronym | Definitions and Coding |
|---|---|---|
| Dependent Variable: Total IR content elements and HEIs core activities Disclosure Score | TOTAL | Total IR Content Elements and HEIs core activities Disclosure Score. Where, TOTAL is the IR content elements and HEIs core disclosure score containing 77 items based on 11 main themes (see Appendix A for more details), including (1) Organisational Overview and External Environment (OEE) including 7 items; (2) Governance (GVN) containing 7 items; (3) Value Creation Model (VCM) covering 7 items; (4) Risk and Opportunity (RO) entailing 7 items; (5) Strategy and Resource Allocation (SRA) including 7 items; (6) Performance (PM) containing 7 items; (7) Outlooks (OLK) covering 7 items; (8) Basis of Preparation and Presentation (BPP) covering 7 items each and these 8 themes re included as IR content elements disclosure (IRED) |
| | | Teaching and Learning (T&L) containing 7 items; Internationalisation (INT) containing 7 items and Research (RSH) containing 7 items. All (11 themes × 7 items) 77 items have a score threshold of 0 to 3, resulting in a total potential score of (77 × 3) 231. Where no disclosure = 0, descriptive disclosure without any link to strategy, governance, performance and prospect = 1, descriptive disclosure and link with all strategy, governance, performance and prospect compare with historic position = 2, descriptive disclosure linked with all strategy, governance, performance and prospect compare with historic, present and future position = 3; |
| Independent Variables related to the Higher Education sector characteristics | RGM | Russell Group Membership: 1, if an HEI member of Russell Group, 0 otherwise; |
| | IRA | Integrated Reporting Framework Adoption: 1, if an HEI adopted IR Framework, 0 otherwise; |
| | LTR | League Table Position Ranking: Measured by performance Position Ranking in Complete University Guide League Table; |
| | TEFR | Teaching Excellence Framework performance ranking: 1, if an HEI ranked gold; 2, if an HEI ranked silver; 3, if an HEI ranked bronze; |
| | KPIR | Key Performance Indicators Reporting: 1, if an HEI disclose KPIs in an annual report, 0 otherwise; |
| | BSIZE | Number of members in HEI governing board; |
| Control Variables | SIZE | Size measured by Total Assets; |
| | IFUND | Funding measured by Percentage of total annual council funding income to total annual income; |
| | LIQD | Liquidity measured by Current Assets divided by Current Liabilities; |
| | LNAGE | Log of HEIs age. HEIs age refers to the date of obtaining degree providing power; |
| | ES | Entry Standard. HEIs entry standard measured by Complete University Guide; |
| | SS | Student Satisfaction. HEIs student satisfaction measured by Complete University Guide; |
| | RQ | Research Quality. HEIs research quality measured by Complete University Guide; |
| | GP | Graduate Prospects. HEIs prospects measured by the Complete University Guide. |

### 8.5. Data Analysis Techniques and Model Specification

The cross-sectional data collected for this study allowed the researchers to perform numerous data analysis techniques to analyse the relationship between HEI specific charac-

teristics, IR content elements and HEI core activities (T&L, INT and RSH) disclosure. Firstly, descriptive statistics of all study variables were carried out and described in Table 2. Secondly, *t*-test and Mann Whitney U tests were employed to discover relationships between IR content elements and HEI core activities disclosure between IR adoption and highlighted in Tables 3 and 4. Thirdly, Pearson's correlation coefficient was analysed through SPSS to investigate the impact of HEIs specific characteristics on IR content elements and HEI core activities disclosure and Variation Inflation Factor (VIF) to find out any potential existence of multicollinearity between dependent and independent variables. The correlation results are presented in Table 5. VIF results did not show any multicollinearity problems between dependent and independent variables. Finally, panel data regression analysis was employed to investigate the relationship between HEI specific characteristics, IR content elements and HEI core activities disclosure. This resulted in four different models. Model 1 presents Pooled OLS for all dependent, independent and control variables. Model 2 demonstrates a Random Effects (RE) Model. Model 3 demonstrates the Weighted Least Square analysis, while Model 4 incorporates dependent lags.

The first two regression models show the relationship between dependent, independent and control variables while the second two regression models were carried out for robustness checks consistent with prior studies (Ntim et al. 2017; Elamer et al. 2017; Elmagrhi et al. 2019; Hassan et al. 2019). The regression model is specified as:

$$TOTALit = \beta_0 + \beta_1 RGM_{it} + \beta_2 IRA_{it} + \beta_3 LTR_{it} + \beta_4 TEFR_{it} + \beta_5 KPIR_{it} + \beta_6 BSIZE_{it} + \beta_7 SIZE_{it} + \beta_8 IFUND_{it} + \beta_9 LIQ_{it} + \beta_{10} LNAGE_{it} + \beta_{11} ES_{it} + \beta_{12} SS_{it} + \beta_{13} RG_{it} + \beta_{14} GP + \varepsilon_{it}$$

(1)

where TOTAL is total IR content elements and HEI core disclosure score. RGM refers to the HEIs Russell Group membership; IRA refers to the IR framework adoption; *LTR* refers to performance position ranking in the league table; TEFR refers to Teaching Excellence Framework performance ranking; *KPIR* refers to key performance indicators reporting; BSIZE refers to number of members in HEI governing board and control variables of total assets depicted as SIZE; percentage of total annual council funding income to total annual income is referred to as IFUND; current assets divided by current liabilities is LIQ; log of HEIs age is referred to as LNAGE; HEIs entry standard is referred to as ES; student satisfaction referred to as SS; RQ refers to research quality and GP refers to graduate prospects. The statistical programmes SPSS and Gretl were used to analyse the research data.

## 9. Results and Discussion

### 9.1. Descriptive Statistics

Table 2 provides a summary of descriptive statistics for all independent variables, control variables and dependent variables. It presents the IR content elements disclosure of T&L, INT and RSH disclosure separately. The descriptive statistics indicate the large variability of IR content elements and HEI core activities disclosure in the HEIs, which aligns with prior research findings (Gordon et al. 2002; Coy and Dixon 2004; Ntim et al. 2017; Hassan et al. 2019). The score ranges from a minimum of 50 to 176 in Total IR content elements and HEI core activities disclosure score, indicating a widespread distribution. Total disclosure relating to T&L from a minimum of 0 to a maximum of 15, INT from a minimum of 0 to a maximum of 16 and RSH from a minimum of 2 to a maximum of 18. In terms of independent variables, total disclosure score related to LTR reveals from 0 to a maximum of 128 and BSIZE intervals from a minimum of 12 to a maximum of 45.

**Table 2.** Descriptive statistics of study variables.

| Variable | Mean | Std. Dev. | Min | Max |
|---|---|---|---|---|
| Dependent Variable | | | | |
| | 99.61 | 20.22 | 50 | 176 |
| TOTAL | | | | |
| IRED (IR content elements disclosure score) | 78.44 | 14.04 | 40 | 130 |
| T&L | 7.11 | 2.62 | 0 | 15 |
| INT | 6.29 | 3.71 | 0 | 16 |
| RSH | 7.76 | 2.43 | 2 | 18 |
| Independent Variables | | | | |
| LTR | 62.57 | 36.93 | 0 | 128 |
| BSIZE | 24.33 | 5.69 | 12 | 45 |
| Control Variables | | | | |
| SIZE | 6,077,468 | 889,419.9 | 34,798 | 7,940,500 |
| IFUND | 31.97 | 343.79 | 0 | 6615 |
| LIQD | 1.92 | 1.1 | 0.21 | 9.08 |
| LNAGE | 1.53 | 0.5 | 0.47 | 2.96 |
| ES | 351.73 | 79.98 | 0 | 601 |
| SS | 4.05 | 0.32 | 0 | 5 |
| RQ | 2.67 | 0.51 | 0 | 3.36 |
| GP | 68.61 | 11.5 | 0 | 95.1 |
| Dichotomous Variables | No (0) | Yes (1)/Gold | | |
| RGM | 297 | | | |
| | | 72 | | |
| IRA | 341 | 28 | | |
| KPIR | 75 | 294 | | |
| TEFR | 36 | 126 | | |

Analysis of Integrated Reporting Content Elements and Higher Education Institutions Core Activities Disclosure Checklist

The researchers carried out two different types of analysis to explain the IR content elements and HEI core activities disclosure checklist. First, the total disclosure scores over the selected three years study period (2015–2016, 2016–2017 and 2017–2018) for all HEIs in our sample presented in Table 3. The results demonstrate that there is an increase in the trend and of IR content elements and HEIs core activities disclosure provided by sample HEIs over the study periods. For instance, the average HEIs disclose 85.24 (36.91%), 97.56 (42.23%) and 117.23 (50.74%), respectively, the disclosure trend increased by 37.46% over the three-year periods investigated, which seems to suggest that UK HEIs have some relationship between IR content elements and core activities disclosure. On the other hand, similar increased patterns can be observed in relation to the four thematic areas. For example, IR content elements (IRED) disclosures are between 40 (23.80%) and 114 (67.85%) with an average of 69.58 (41.41%) disclosure checklist increasing steadily from 2015 to 2016 and 2017 to 2018. T&L disclosures are between 3 (14.28%) and 13 (61.90%) with an average of 5.99 (28.52%) on the disclosure checklist, increasing in similar patterns with IR content elements disclosure.

These results indicate that HEI core activities (T&L, INT and RSH) disclosure patterns are approximating IR content elements disclosure. Therefore, this study argues that the increase of IR content elements and HEI core activities disclosure suggests that HEIs are attracted to the IR framework and have started to implement an IT approach. As IT approach has two components, one is connected with the organizational strategy, governance, past performance and future prospects, and the other is connectivity and interdependencies between the factors that have significant effect on value creation as an organizational cultural (Dumay and Dai 2017). Due to this fact, if organizations are engaged actively with considering the relationship between operating and financial departments then they commence to produce the integrated report by employing holistic approach across the organization (Al-Htaybat and Alberti-Alhtaybat 2018). Therefore, the company can only prepare integrated report after implementing the IT approach for their effective communication to finalize with financial and non-financial information need to be reported to tackle stakeholders' demands.

This is arguably due to the IR content elements categories (OEE, GOV, VCM, RO, SRA, PM, OLT, BPP) and HEI core activities (T&L, INT and RSH) uniting together information from various departments of HEIs, which confirms the interrelationship and interdependencies as a reflection of IT. IT can break down the different departmental barriers through corporate dialogue between teams during the process of corporate report preparation (Adams 2017; Hassan et al. 2019).

**Table 3.** Relationship between integrated reporting content elements and HEIs core activities and IR adoption for the study periods.

| Variables | Total Sample (123) | | | IR Adopted HEIs (12) | | | IR not Adopted HEIs (111) | | |
|---|---|---|---|---|---|---|---|---|---|
| | 2015/16 | 2016/17 | 2017/18 | 2015/16 | 2016/17 | 2017/18 | 2015/16 | 2016/17 | 2017/18 |
| Total IR content elements and HEIs core activities disclosure Score | | | | | | | | | |
| Mean | 85.24 | 97.56 | 117.23 | 112.18 | 128.42 | 146.17 | 84.77 | 96.53 | 107.92 |
| STD | 14.27 | 16.41 | 17.94 | 23.54 | 24.53 | 26.07 | 12.93 | 14.17 | 13.12 |
| Min | 50 | 63 | 83 | 76 | 86 | 100 | 50 | 70 | 81 |
| Max | 138 | 159 | 181 | 151 | 162 | 176 | 138 | 150 | 170 |
| IR content elements disclosure (IRED) | | | | | | | | | |
| Mean | 69.58 | 78.55 | 87.2 | 85.73 | 97.92 | 111.25 | 67.78 | 76.46 | 84.59 |
| STD | 11.16 | 12.29 | 12.74 | 15.3 | 15.89 | 16.75 | 9.13 | 9.84 | 9.06 |
| Min | 40 | 56 | 70 | 61 | 69 | 79 | 40 | 56 | 70 |
| Max | 114 | 120 | 130 | 114 | 120 | 130 | 101 | 108 | 123 |
| Teaching and Learning disclosure (T&L) | | | | | | | | | |
| Mean | 5.99 | 7.11 | 8.24 | 8.73 | 10.17 | 11.67 | 5.67 | 6.77 | 7.86 |
| STD | 2.5 | 2.48 | 2.39 | 3 | 2.91 | 3.14 | 2.22 | 2.21 | 1.98 |
| Min | 3 | 3 | 5 | 3 | 6 | 7 | 0 | 0 | 3 |
| Max | 13 | 13 | 15 | 13 | 13 | 15 | 10 | 12 | 15 |
| Internationalisation disclosure (INT) | | | | | | | | | |
| Mean | 4.99 | 6.28 | 7.61 | 8 | 9.33 | 11 | 4.65 | 5.95 | 7.24 |
| STD | 3.53 | 3.84 | 3.3 | 3.92 | 4 | 3.54 | 3.35 | 3.69 | 3.07 |
| Min | 0 | 0 | 0 | 0 | 0 | 4 | 0 | 0 | 0 |
| Max | 16 | 16 | 16 | 12 | 13 | 15 | 16 | 16 | 16 |
| Research disclosure (RSH) | | | | | | | | | |
| Mean | 6.98 | 7.71 | 8.61 | 9.73 | 11 | 12.25 | 6.68 | 7.35 | 8.22 |
| STD | 2.21 | 2.36 | 2.43 | 3.1 | 3.41 | 3.79 | 1.9 | 1.93 | 1.88 |
| Min | 2 | 2 | 3 | 2 | 5 | 6 | 2 | 2 | 3 |
| Max | 15 | 16 | 18 | 15 | 16 | 18 | 11 | 14 | 16 |

Similarly, Table 3 presents the analogous increasing patterns in IR content elements and HEI core activities disclosure in both IR adopted and non-IR adopted HEIs. Even though very limited number of HEIs adopted IR, the UK HEIs are strongly encouraged to disclosure more activities, which have done for their long term, value creation for their stakeholders as a corporate citizen (Adams 2018; King Report IV 2016). Likewise, BUFDG (2016, 2017) called UK HEIs to take part in an IR pilot study and six universities participated (BUFDG 2016). Therefore, HEIs may be encouraged to adopt IR frameworks voluntarily to legitimise themselves and fulfil social expectation (Adams et al. 2016).

However, HEIs that have adopted IR have slightly higher IR content elements and core activities disclosure in comparison with their non-IR adopted counterparts. The average IR adopted HEIs scored 112.18 (48.56%) of the total of 231 points in 2015–16, with 128.42 (55.59%), and 146.17 (63.27%) in 2016–2017 and 2017–2018, respectively. In terms of HEI core activities disclosure, the average IR adopted HEIs scored 26.46 (42%) of the total of 63 points in 2015–16, with 30.5 (48.41%) and 34.92 (55.42%) in 2016–0217 and 2017–2018, respectively.

The researchers carried out *t*-tests and Mann-Witney U tests to explore if there were any differences in the disclosure trend of IR content elements and HEI core activities and the IR framework adoption within UK HEIs (IR framework adopted, or non-IR framework adopted). Table 4 presents the totals of all IR content elements and HEI core activities disclosure.

**Table 4.** Relationship between IR content elements, teaching and learning, internationalisation and research activities disclosure and IR framework adoption.

| | Linking Disclosure Items' Categories to IR Framework Adoption of HEI (IRA) | | | | | | | | | | | |
| | IR adopted HEIs (12) | | | | IR Not Adopted HEIs (111) | | | | *t* Test | | Mann-Whitney U Test | |
| | Mean | St | Min | Max | Mean | St | Min | Max | *t* Test | *p*-Value | Z | *p*-Value |
|---|---|---|---|---|---|---|---|---|---|---|---|---|
| (1) IR content elements disclosure (IRED) | 98.44 | 18.52 | 61 | 130 | 76.28 | 11.58 | 40 | 123 | −7.031 | 0.000 *** | −6.7 | 0.000 *** |
| (2) Teaching and Learning disclosure (T&L) | 10.28 | 3.14 | 5 | 15 | 6.77 | 2.31 | 0 | 15 | −6.515 | 0.001 ** | −5.841 | 0.000 *** |
| (3) Internationalisation disclosure (INT) | 9.5 | 3.85 | 0 | 15 | 5.95 | 3.53 | 0 | 16 | −5.292 | 0.996 | −5.509 | 0.000 ** |
| (4) Research disclosure (RSH) | 11 | 3.46 | 5 | 18 | 7.41 | 2 | 2 | 16 | −6.101 | 0.000 *** | −6.149 | 0.000 *** |
| (5) Total IR content elements and HEIs core activities Disclosure score | 129.22 | 27.42 | 76 | 176 | 96.41 | 16.39 | 50 | 170 | −7.044 | 0.000 *** | −6.664 | 0.000 *** |

Significance level: ** $p < 0.05$, *** $p < 0.01$.

The findings indicate that, in general, IR adopted HEIs have more disclosure trend on IR content elements and core activities. *t*-test test identified significant differences (*t*-test $p = 0.000$ and Mann-Whitney $p = 0.000$) between IR adopted institutions and non-IR adopted institutions with regards to most of all four themes. For example, IRED (*t*-test $p = 0.000$ and Mann-Whitney $p = 0.000$), T&L (*t*-test $p = 0.001$ and Mann-Whitney $p = 0.000$), INT (*t*-test $p = 0.996$ and Mann-Whitney $p = 0.000$), and RSH (*t*-test $p = 0.000$ and Mann-Whitney $p = 0.000$). Collectively, the above results indicate that HEIs that adopt IR provide a higher level of IR content elements and core activities disclosure in comparison with their non-IR adopted counterparts. This supports H2: There is a positive relationship between IR framework adopted universities and the level of IR content elements, teaching and learning, internationalisation and research activities disclosure.

*9.2. Correlation Matrix*

Table 5 presents the correlation matrix for all variables employed in the regression analysis to test for multicollinearity and bivariate Pearson parametric coefficient was carried out. The results show that there was a positive, though not significant, relationship between IR content elements and HEI core activities disclosure and RGM (0.083), KPIR (0.093) and BSIZE (0.031). However, a negative and significant relationship was observed between TEFR (−0.162) and negative though not significant relationship found between LTR (−0.062) on IR content elements and HEI core activities disclosure. In terms of control variables, the results reveal that there was a positive and significant relationship between SIZE (0.140), LIQ (0.120), LNAGE (0.171), ES (0.146) and GP (0.177) on IR content elements and HEI core activities disclosure. In addition, the correlation amongst the variables was relatively low, suggesting that there were no serious multicollinearity problems. The multicollinearity problem is present if the correlation among variables is high (generally 0.90 and above) (Hair et al. 2013; Hassan et al. 2019). The multicollinearity appeared to be problematic if the VIF for any variable of the research was more than 10, or the tolerance of any variable was less than 0.1 (Gujarati 2003). Therefore the researchers carried out VIF tests and the results of all variables were less than 6, confirming that there were no multicollinearity problems.

**Table 5.** Correlation matrix for research variables.

| | Total | RGM | IRA | LTR | TEFR | KPIR | BSIZE | SIZE | IFUND | LIQ | LNAGE | ES | SS | RQ | GP |
|---|---|---|---|---|---|---|---|---|---|---|---|---|---|---|---|
| Total | 1 | | | | | | | | | | | | | | |
| RGM | 0.083 | 1 | | | | | | | | | | | | | |
| IRA | 0.522 ** | 0.147 ** | 1 | | | | | | | | | | | | |
| LTR | −0.062 | −0.586 ** | −0.004 | 1 | | | | | | | | | | | |
| TEFR | −0.162 ** | −0.102 * | 0.002 | 0.317 ** | 1 | | | | | | | | | | |
| KPIR | 0.093 | −0.283 ** | 0.086 | 0.178 ** | 0.049 | 1 | | | | | | | | | |
| BSIZE | 0.031 | 0.197 ** | 0.031 | −0.324 ** | −0.197 ** | −0.074 | 1 | | | | | | | | |
| SIZE | 0.140 ** | 0.635 ** | 0.054 | −0.498 ** | −0.163 ** | −0.181 ** | 0.131 * | 1 | | | | | | | |
| IFUND | 0.001 | −0.025 | −0.011 | 0.076 | −0.117 * | 0.026 | 0.021 | −0.023 | 1 | | | | | | |
| LIQ | 0.120 * | −0.087 | 0.230 ** | 0.166 ** | 0.031 | 0.131 * | −0.07 | −0.024 | −0.04 | 1 | | | | | |
| LNAGE | 0.171 ** | 0.514 ** | 0.087 | −0.659 ** | −0.184 ** | −0.045 | 0.256 ** | 0.578 ** | −0.012 | −0.191 ** | 1 | | | | |
| ES | 0.146 ** | 0.039 | 0.04 | −0.746 ** | −0.206 ** | −0.195 ** | 0.205 ** | 0.649 ** | −0.03 | −0.221 ** | 0.629 ** | 1 | | | |
| SS | 0.065 | 0.039 | 0.007 | 0.028 | 0.041 | −0.056 | −0.147 ** | 0.062 | −0.017 | −0.049 | −0.03 | 0.355 ** | 1 | | |
| RQ | 0.098 | 0.479 ** | 0.017 | −0.645 ** | −0.121 * | −0.145 ** | 0.139 ** | 0.420 ** | −0.029 | −0.169 ** | 0.604 ** | 0.695 ** | 0.398 ** | 1 | |
| GP | 0.177 ** | 0.510 ** | −0.013 | −0.635 ** | −0.172 ** | −0.164 ** | 0.122 * | 0.443 ** | −0.047 | −0.191 ** | 0.456 ** | 0.769 ** | 0.486 ** | 0.620 ** | 1 |

Notes: the above table contains Pearson's Parametric correlation coefficients, significance level: * $p < 0.05$; ** $p < 0.01$. Variables are defined as per Table 1.

*9.3. Multivariate Results*

Table 6 presents the regression results for total IR content elements and HEI core activities disclosure and all independent and control variables.

**Table 6.** Influence of HEI specific characteristics on IR content elements and core activities disclosure.

| Variables | Pooled OLS (1) | (2) RE | (3) WLS | (4) Dependent Lags |
|---|---|---|---|---|
| | Panel A: Independent variables | | | |
| | z-value *p*-value | z-value *p*-value | z-value *p*-value | z-value *p*-value |
| RGM | −1.056 ** (0.011) | −0.723 (0.877) | −3.196 ** (0.022) | −1.306 (0.149) |
| IRA | 6.626 *** (0.000) | 4.181 *** (0.000) | 15.47 *** (0.000) | 5.122 *** (0.000) |
| LTR | 3.680 *** (0.008) | 0.781 (0.685) | 2.909 *** (0.000) | 2.117 *** (0.009) |
| TEFR | −3.677 *** (0.000) | −2.080 ** (0.021) | −4.615 *** (0.000) | −0.258 (0.159) |
| KPIR | 3.982 *** (0.001) | 2.588 *** (0.004) | 1.923 *** (0.000) | 0.350 (0.674) |
| BSIZE | 0.578 (0.949) | 0.134 (0.803) | 0.437 (0.870) | 0.027 (0.972) |
| | Panel B: Control variables | | | |
| SIZE | 2.200 ** (0.060) | 1.818 ** (0.035) | 1.401 ** (0.045) | 0.946 (0.183) |
| IFUND | 0.328 (0.794) | 0.295 (0.119) | 0.222 (0.827) | 0.264 (0.606) |
| LIQ | 0.519 (0.447) | 0.386 (0.943) | 0.525 (0.919) | 0.096 (0.809) |
| LNAGE | 0.354 (0.135) | 0.143 (0.619) | 1.489 (0.562) | 0.671 (0.202) |
| ES | 0.176 (0.031) | 0.147 (0.807) | 0.424 (0.167) | 0.003 (0.703) |
| SS | −2.747 *** (0.003) | −1.09133 | −1.693 *** (0.001) | −0.293 (0.664) |
| RQ | 2.808 *** (0.002) | 0.950 (0.306) | 0.331 *** (0.001) | 1.260 * (0.053) |
| GP | 2.172 ** (0.015) | 1.070 (0.328) | 1.620 ** (0.011) | 0.186 (0.162) |
| _cons | −15.13 *** (0.000) | −18.89 *** (0.000) | −12.17 *** (0.000) | −8.158 (0.617) |
| Years | Included | Included | Included | Included |
| N | 369 | 369 | 369 | 246 |
| R-sq | 0.57 | N/A | 0.56 | 0.91 |
| adj. R-sq | 0.55 | N/A | 0.54 | 0.91 |

Notes: the above table represents regression coefficients and *p*-value in parentheses. Significance levels are * $p < 0.10$, ** $p < 0.05$, *** $p < 0.01$. The variables are defined as per Table 1.

Russell Group Membership and Integrated Reporting Content Elements and Higher
Education Institutions Core Activities Disclosure

Model 1 of Table 6 demonstrates that the RGM of HEIs was negatively and significantly
associated with IR content elements and core activities disclosure ($t = -1.056$). This
supports the argument that the RGM HEIs focus on research, obtain higher research grants
and have lower voluntary disclosure (Pursglove and Simpson 2007). The newer universities
are far less well-resourced, and teaching-led universities have higher student satisfaction
and use higher voluntary disclosure to attract talented students in comparison with RGM
universities (Boliver 2015). However, the results are not consistent with Ntim et al. (2017)
who did not find any significant relationship between RGM and voluntary disclosure in
UK HEIs. Melloni et al. (2017) concluded that the corporations who have a higher financial
position will produce more concise, readable and comprehensive integrated reports.

Our results indicate that there is a significant and negative relationship between
RGM and IR content elements and core activities disclosure. Thus, we reject H1: There
is a positive relationship between Russell Group member universities and the level of
IR content elements, teaching and learning, internationalisation and research activities
disclosure. This may be because very few universities have started to publish integrated
reports, and this is the first study to explore IR content elements and HEI core activities
disclosure.

*9.4. Integrated Reporting Framework Adoption and Total Integrated Reporting Content Elements
and Higher Education Institutions Core Activities Disclosure*

Model 1 of Table 6 indicates that there is a positive significant association between
IR framework Adoption (IRA) and the IR content elements and HEI core activities disclo-
sure ($t = 6.626$). This result is in line with numerous prior studies which find a positive
relationship between IR reporting framework adoption and total IR disclosure (Melloni
et al. 2017; Pavlopoulos et al. 2017; Hassan et al. 2019). Additionally, this research supports
the previous study by Feng et al. (2017) who suggested IR framework is well developed
to drive organisational activities reflection through corporate report for all profit-oriented
and non-profit oriented organisations. Therefore, our results provide support to H2: There
is a positive relationship between IR framework adopted universities and the level of
IR content elements, teaching and learning, internationalisation and research activities
disclosure.

Additionally, our results suggest that those universities who adopted the IR framework
employed the IT approach to collect the required information. This enabled the prepared
external reports to legitimise their strategy and to differentiate themselves from other
HEIs demonstrating to their stakeholders how they create value for short, medium and
long-term periods (Pavlopoulos et al. 2017; Rinaldi et al. 2018; Adams 2018).

League Table Ranking Position and Integrated Reporting Content Elements and Higher
Education Institutions Core Activities Disclosure

Model 1 of Table 6 demonstrates the positive and significant relationship between
LTR and IR content elements and HEI core activities disclosure ($t = 3.680$). This result
is consistent with previous research which finds that the university ranking position is
the result of strategic decisions concerning voluntary disclosure and HEI marketing
(Luca and Smith 2015). Higher-ranked universities offer a more complete picture of RSH
performance, can be supportive to the stakeholders (Robinson-Garcia et al. 2014). LTR is
crucial to the HEIs to convey the quality of their activities and they prefer more disclosure
if they have higher ranking (Christie 2017). Therefore, the research findings support H3:
There is a positive relationship between the league table ranking position and the level
of IR content elements, teaching and learning, internationalisation and research activities
disclosure. However, these findings are not in line with the research findings explored
by (Gibbons et al. 2015) who found the year-to-year movement of institutional ranking do
not necessarily represent real changes. Students are not strongly influenced by annual

university ranking changes; however, their decision is associated with the quality of the learning environment.

### 9.5. Teaching Excellence Framework Performance and Integrated Reporting Content Elements and Core Activities Disclosure

Model 1 of Table 6 shows a negative and significant influence of TEFR and the level of IR content elements and core activities disclosure ($t = -3.677$). This result is in line with prior studies which find that it is challenging to assess university teaching function and their research outcomes in the same way and requires more empirical studies (Wild and Berger 2016). However, this result is not consistent with Barkas et al. (2019) which find that TEF appears to be positive, however, they also find that implementation is problematic and needs to be supported by evidence-based research. Therefore, our results do not support a positive relationship between TEFR and IR content elements and HEI core activities disclosure. Thus, we reject H4: There is a positive relationship between the teaching excellence framework ranking position and the level of IR content elements, teaching and learning, internationalisation and research activities disclosure. The results may be due to an early stage of *TEF* implementation with UK HEIs. In addition, this result is in line with Hassan et al. (2019) who suggest that the university policymakers, such as BUFDG, may consider developing voluntary IR guidelines to produce clear, concise and comprehensive corporate reports where the HEI can link their IT approach to do more voluntary and core activities disclosure.

### 9.6. Key Performance Indicators Reporting and Integrated Reporting Content Elements and Core Activities Disclosure

Model 1 of Table 6 displays the regression results of KPIR and IR content elements and core activities disclosure. Our results show that there a positive significant relationship between KPIR and IR content elements and HEI core activities disclosure ($t = 3.982$). This is consistent with a legitimacy argument in that universities that disclose more information are seen as more legitimate within society and fulfil stakeholder expectations (Maingot and Zeghal 2008). This result is in line with the findings of prior research from various countries. In a Canadian context, researchers found that the larger size universities who engaged in KPIR have higher voluntary disclosure (Maingot and Zeghal 2008). KPIs are a crucial element in justifying teaching, research, supervision, publication, consultancy and universities disclose all activities to their stakeholders in Malaysia (Masron et al. 2012). Each public university should report KPIs mainly to measure the number of graduates, research outputs, academic services, university vision, mission and disclose all results to the stakeholders in a Thailand university context (Sukboonyasatit et al. 2011). Therefore, our research findings support H5: There is a positive relationship between KPIR and the level of IR content elements, teaching, and learning, internationalisation, and research activities disclosure.

### 9.7. University Governing Board Size and Integrated Reporting Content Elements and Core Activities Disclosure

Model 1 in Table 6 indicates that there is no significant relationship between BSIZE and IR content elements and HEI core activities disclosure ($t = 0.578$). This finding is consistent with various prior research findings which found no relationship between BSIZE and voluntary disclosure (Gallego-Alvarez et al. 2011; Ntim et al. 2017; Hassan et al. 2019). Additionally, these findings are in line with Hassan et al. (2019) recommendation which suggested that further governance may lead to higher demand for IR content elements and HEI core activities disclosure to maintain public accountability and preserve legitimacy. Therefore, our results do not support the positive relationship between BSIZE and IR content elements and HEI core activities disclosure. Thus, we reject H6: There is a positive relationship between university governing board size and the level of disclosure on IR content elements, teaching and learning, internationalisation and research disclosure. In this context, our study population may not have enough variables to judge voluntary

disclosure. The researchers suggest more empirical research with more governance-related variables, such as board meetings, directors pay, gender, duality and other appropriate items as this may provide different results.

## 10. Additional Analysis

Additional tests were carried out to determine the robustness of the main results from the previous section. First, the Hausman test was carried out and the results revealed that $p > 0.05$. Therefore, we use RE regression analysis (Ntim et al. 2017; Elamer et al. 2017; Hassan et al. 2019) to investigate whether HEI specific characteristics have any influence on IR content elements and core activities disclosure. Omitted variables are potential sources of endogeneity in this research context; HEIs with some specific features could choose to disclose more IR content elements and core activities related information. Reverse causality is another potential source of endogeneity. In this context, the result of the Model of Table 6 could be biased. To deal with endogeneity, the researchers use a RE regression as follows:

$$\text{TOTAl} = \beta_0 + \beta_1 \text{RGM} + \beta_2 \text{IRA} + \beta_3 \text{LTR} + \beta_4 \text{TEFR} + \beta_5 \text{KPIR} + \beta_6 \text{BSIZE} + \beta_7 \text{SIZE} + \beta_8 \text{IFUND} + \beta_9 \text{LIQ} + \beta_{10} \text{LNAGE} + \beta_{11} \text{ES} + \beta_{12} \text{SS} + \beta_{13} \text{RG} + \beta_{14} \text{GP} + \delta_{it} + \varepsilon_{it} \quad (2)$$

Everything else remains unaffected as stated in Equation (2) and Table 1. $\delta$ is the university-year specific effects, and $\varepsilon$ is the error term. The results are displayed in Model 2 of Table 6. These results are highly similar to those represented in Model 1 of Table 6, suggesting that our results seem to be robust to the potential endogeneities that may be affected by omitted variable bias and/or reverse causality.

Second, the researchers further addressed potential endogeneities that might be affected by omitted variable bias by estimating using weighted least squares (Aljifri 2008; Shi et al. 2012; Lang and Sul 2014). The researchers use the weighted variable of the BSIZE and re-run Equation (2) as follows:

$$\text{TOTAl} = \beta_0 + \beta_1 \text{RGM} + \beta_2 \text{IRA} + \beta_3 \text{LTR} + \beta_4 \text{TEFR} + \beta_5 \text{KPIR} + \beta_6 \text{BSIZE} + \beta_7 \text{SIZE} + \beta_8 \text{IFUND} + \beta_9 \text{LIQ} + \beta_{10} \text{LNAGE} + \beta_{11} \text{ES} + \beta_{12} \text{SS} + \beta_{13} \text{RG} + \beta_{14} \text{GP} + \delta_{it} + \varepsilon_{it} \quad (3)$$

Everything else remains unaffected as stated in Equation (2) except that the researchers use the weighted part of the BSIZE. The results are reported in Model 3 of Table 6. These results are also similar to those presented in Model 1 of Table 6, indicating that our findings appear to be robust to potential endogeneity that may be caused by omitted variables bias.

Third, to ascertain the assumption underlying our OLS regression model that all the unobserved heterogeneities may affect the correlation between the HEIs sector-specific variables and the error term is invariable over time, the researchers calculate a lagged estimator as proposed by Ntim et al. (2017) and Hassan et al. (2019). The findings are reported in Model 4 of Table 6. The results indicate a negative though not statistically significant relationship amongst the RGM, TEFR and TOTAL indices and a positive and statistically significant relationship among the IRA, LTR and TOTAL indices and a positive but not statistically significant relationship among the KPIR, BSIZE and TOTAL indices. These results are largely similar to those reported in Model 1 of Table 6 except the significance on RGM and KPIR though the effect remains the same thereby implying that our results are not strongly affected by potential endogeneity problems that may be caused by simultaneity.

## 11. Conclusions

Despite the wide range and significant impact of the activities undertaken at universities, they have tended to lag the rest of the corporate world when it comes to identifying and communicating their activities and impacts to the diverse groups of stakeholders that are involved in their existence. To bridge the gap between stakeholder expectations and organisational communication style regarding transparency and conciseness the IR framework was developed (IIRC 2013). The main motivation for this study is that the unique nature of HEIs challenges makes the connection and the interdependence between

its departments crucial to provide relevant information to stakeholders about their value creation process. The researchers conceptualise IT as an internal process that organisations can follow to increase the level of disclosure on IR content elements and HEI core activities, which will be used as a communication tool with stakeholders, as the organisation needs to clearly communicate with all departments from operating and management level to collect financial and non-financial information to produce an integrated report. As a result, organisations can disclose their strategy, governance, business model, resources allocation procedure, performance for their long-term value creation through the integrated report. In doing so, this research contributes to the link between IR and IT research (Katsikas et al. 2017; Adams 2017; Rinaldi et al. 2018) by investigating whether the disclosure of IR content elements reflects the implementation of an IT approach in HEIs (Higgins et al. 2019; Adams 2018; Hassan et al. 2019). This research introduces legitimacy theory to describe the strategic thinking of HEIs; that communication via IT can close the gap between the organisation and its stakeholders and enhance its credibility (Maingot and Zeghal 2008; Veltri and Teresa Nardo 2013b; Rinaldi et al. 2018). This might enable HEIs to 'live their story, rather than merely telling it' (CIMA 2017; BUFDG 2017).

Our results support the idea that IT is contributing to the enhancement of the level of disclosure on IR content elements and core activities in HEIs. The results of this study are in line with prior studies that suggest that the IIRC's success is due to its ability to take advantage of a favourable momentum, appearing when corporate reporting was already beginning to become more integrated in practice before issuing the IR framework (Adams et al. 2016; La Torre et al. 2019). Our results, when measuring the relationship between IR content elements and HEI core activities disclosure and IR adoption, indicated that universities can improve their disclosure through IT and legitimise their strategy even without adopting the IR framework.

Using recent data from three academic periods for 123 HEIs in the UK, the findings indicate that the trends of IR content elements and HEI core activities disclosure were increasing during the study period. Analysis over the three academic periods of 2015–2016, 2016–2017 and 2017–2018 with independent variables and control variables produced average scores of 36.90% in 2015–2016, increased to 50.74% in 2017–2018 with all independent variables though without the inclusion of the control variables. The overall score was 43.12%. This appears to be very low in comparison with other studies, such as voluntary disclosure in HEI sector 44.02% from 130 UK HEIs (Ntim et al. 2017), and 56.9% from 78 Spanish HEIs (Gallego-Alvarez et al. 2011). However, this study is in line with IR content elements disclosure of 34.40% in 135 UK HEIs by Hassan et al. (2019). Akin to the business organisational sector, the lack of IR content element disclosure could be due to the HEIs lack of expertise and/or lack of resources to produce an integrated report appropriately. This study focussed on HEI annual report disclosure and did not consider the possibility that HEIs may rely more on other forms of public communication (website, press release, social media). From a methodological point of view, the disclosure index was constructed based on the IR framework produced by the IIRC (2013), PAI produced by Coy and Dixon (2004) and the INT index produced by Elkin et al. (2005). However, the IR framework implication was still at an early stage and still requires a lot of dialogue to support implementation in the HEIs sector (Veltri and Silvestri 2015; Brusca et al. 2018; Hassan et al. 2019). In the UK HEIs sector, professional bodies are actively engaged to support IR framework adoption and integrated report preparation (BUFDG 2016).

These findings have important policy, regulatory, managerial and international implications. Firstly, the results will be of interest to policymakers and regulators to assess the benefits of adopting IT on increasing the level of disclosure on IR content elements and HEI core activities disclosure. This might provide evidence for the possibility of the mandatory implementation of IIRC guidelines. Secondly, our results will be of interest to managers at universities that wish to follow these new trends. The findings can serve as a learning process for institutions interested in implementing IT that leads to better corporate reporting. Thirdly, our results are important to other stakeholders, such as

potential students, academics and other stakeholders to encourage universities to address IR issues in their reporting as this will increase their impact on society.

This study has some limitations. The use of the weighted index may need more simplification and may be affected by judgement in the selection of content; however, it has been used before (Haji and Anifowose 2017; Adhikariparajuli et al. 2021). Future research can focus on using an unweighted index and compare the results. The study is based on IR content elements only and could be extended to include the fundamental concept and basic principles of the IR framework. This research looks at other aspects of the IR framework, such as reporting guidelines and HEI core activities (T&L, INT and RSH), and further research can investigate community engagement, other charity work, environmental activities, mental health and sports-related activities which have been omitted from this study. These factors should be examined in more depth by future researchers both nationally and internationally which could extend the research findings. The study focused on one country (UK), future research should extend our work to be examined internationally and it might be more interesting to compare voluntary disclosures of IR to HEI disclosure in other countries where IR is mandatory, such as South Africa.

**Author Contributions:** Conceptualization, M.A.; A.H. and M.F.; methodology, M.A. and A.H.; software, M.A.; validation, M.A.; A.H. and M.F.; formal analysis, M.A. and A.H.; investigation, M.A.; A.H. and M.F.; resources, M.A. and A.H.; data curation, M.A. and A.H.; writing—original draft preparation, M.A. and A.H.; writing—review and editing, M.A. and A.H.; visualisation, M.A. and A.H.; supervision, A.H. and M.F.; project administration, M.A. and A.H.; funding acquisition, A.H. All authors have read and agreed to the published version of the manuscript.

**Funding:** This research received no external funding.

**Institutional Review Board Statement:** Not applicable.

**Informed Consent Statement:** Not applicable.

**Conflicts of Interest:** The authors declare no conflict of interest.

## Appendix A

| Items of Organisational Overview and External Environment (OEE) Disclosure | 0 | 1 | 2 | 3 |
|---|---|---|---|---|
| Vision & Mission (VM) | | | | |
| Operating activities & Principle Activities (OPA) | | | | |
| Competitive Environment and Institutions Position (CEIP) | | | | |
| Quantitative information about the Number of Student and number of Staff (NSS) | | | | |
| Commercial, Social, Technical and Environmental (CSTE) (which significantly affect HEIs) | | | | |
| Revenue from Various Activities and any changes from the prior year (RVA) | | | | |
| Information of commercial, social, technical and Environmental factors' Impact on Value Creation (EIVC) | | | | |
| **Items of Governance (GOV) Disclosure** | | | | |
| Leadership structure, Skill and Diversity of Governors (LSDG) | | | | |
| Role of HEIs' Different Leadership Structure and their interaction (DLS) | | | | |
| Role and responsibilities of Executive and Non-executive Members (RENM) | | | | |
| Strategic Decision-Making Process within the HEIs (SDMP) | | | | |
| Governor Risk Monitoring Actions (RMA) | | | | |
| Explain how Governors Mitigate the Risk (RMR) | | | | |
| How the Leadership Team Members Remunerated (LTMR) | | | | |

| Items of Value Creation Model (VCM) Disclosure | | | | |
|---|---|---|---|---|
| Value Creation Model for HEIs to fulfil their strategic purpose and value creation for short, medium and long-term (VCMH) | | | | |
| How the HEIs Main Activities draw the financial, intellectual, human and natural Resources (MAR) | | | | |
| HEIs' Main Income Source over time (MIS) | | | | |
| Connection between Value Creation Model and Key Performance Indicators (VCMKPI) | | | | |
| Social and Environmental Impact of HEIs activities including positive and negative (SEI) | | | | |
| Information about Staff Satisfaction (SS) | | | | |
| Procedure of Organisational Change adoption, Staff training, development and improvement (OCSD) | | | | |
| **Items of Risk and Opportunity (RO) Disclosure** | | | | |
| Institution's approach to Identify Risks and Opportunities, define "significant" risk and opportunity (IRO) | | | | |
| Set out the Significant Risks that affect HEIs ability to create value over the period (SSR) | | | | |
| Explain the institutions Remaining Risk Management (RRM) | | | | |
| Set out the Significant Opportunities which support Value Creation over the period (SOVC) | | | | |
| How the institution Realises the Opportunities and Anticipated Benefits of the institutions' efforts (ROAB) | | | | |
| Risk Reporting Process (RRP) | | | | |
| Source of risk and affordability to reduce risk and gain the opportunity (SOR) | | | | |
| **Items of Strategy and Resource Allocation (SRA) Disclosure** | | | | |
| Institutions short, medium and long-term Strategic Objectives (ISO) | | | | |
| The Strategy in Place, or Intends to Implement, to achieve its strategic objectives (SPII) | | | | |
| Resource Allocation Plan to support the implementation of its strategy (RAP) | | | | |
| How the institution will ensure its Financial Sustainability Management for short, medium and long-term (FSM) | | | | |
| How the Institution will Measure its Performance (IPM) | | | | |
| How the Institution will Differentiate itself and its Reflection in Strategy, structure and activities (IDRS) | | | | |
| Intellectual Capital Utilisation for revenue maximization (ICU) | | | | |
| **Items of Performance (PM) Disclosure** | | | | |
| Set out Institutions Objectives for the period and the extent which has Achieved them (IOA) | | | | |
| Balanced and complete view of the Institutions Financial Performance (IFP) | | | | |
| Institutions performance about Strategic, Financial, and Human Issue (SFHI) | | | | |
| Impact of institutions Activities towards financial, intellectual, human, and Natural Resources (IAR) | | | | |
| Relationship between Key Stakeholders and institutional response towards their Needs and interests (KSN) | | | | |
| Linkage between Past and Current Performance and Institutions Outlook (LPCPIO) | | | | |
| Institutions activities towards Carbon Emissions and Environmental Sustainability (CEES) | | | | |
| **Items of Outlook (OL) Disclosure** | | | | |
| Institutions Expectation about external Environment likely to face short, medium and long-term (IEE) | | | | |
| The Impact of External Environment towards institutions financial, intellectual, and Natural Resources (IEER) | | | | |
| Institutions Response towards critical Challenges and Uncertainties likely to arise (RCU) | | | | |
| Any Ability to Respond Effectively to external Environmental change (AREE) | | | | |
| Legal and Regulatory requirement to Comply and its Impact on an institution's activities (LRCI) | | | | |
| Any Plan to Respond Uncertainty for short, medium and long-term (PRU) | | | | |
| The interrelationship between Objectives recognised External Sources and any assumption; projection has made (IOES) | | | | |
| **Items of Basis of Preparation and Presentation (BPP) Disclosure** | | | | |
| Corporate Report Content determination process and the People Involved for report preparation (RCPI) | | | | |
| Individuals Reviewed and Approved the Report before its Publication (RARP) | | | | |
| Any Framework and method used to quantify Material Matters included in the report (FMM) | | | | |
| Any Uncertainty about any information which is used for Report Preparation (URP) | | | | |
| Significant Matters Identification that Impact HEIs ability to create value (SMII) | | | | |
| Any Stakeholder Engagement for Materiality determination and Prioritisation (SEMP) | | | | |
| Any issues that affect the reader's understanding of HEIs Strategy, Governance, performance and Prospects (ISGP) | | | | |

| **Items of Teaching and Learning (T&L) Disclosure** | | | | |
| --- | --- | --- | --- | --- |
| Quality of Teaching and Student Satisfaction (QTSS) | | | | |
| Departmental Review and course Completion rate (DRC) | | | | |
| Library service and Online Resources (LOR) | | | | |
| Teaching Service Process (TSP) | | | | |
| Number of Graduates and graduate Employment (NGE) | | | | |
| Computer Service Facilities (CSF) | | | | |
| Student Progress Evaluation (SPE) | | | | |
| **Items of Internationalisation (INT) Disclosure** | | | | |
| International Focus Of Programme (IFOP) | | | | |
| Institutional International Link (IIL) | | | | |
| Student Exchange Programme (SEP) | | | | |
| International Research Collaboration (IRC) | | | | |
| International Department Establishment (IDE) | | | | |
| Staff Interaction in International Context (SIIC) | | | | |
| International Conference Attendance (ICA) | | | | |
| **Items of Research (RSH) Disclosure** | | | | |
| Graduate Pass Rate (GPR) | | | | |
| Number of Research students and Scholarship Scheme (RSS) | | | | |
| All publication in Present and plan for Future Publication (PFP) | | | | |
| Destination of Research Graduate (DRG) | | | | |
| Knowledge and Skills Exchange (KSE) | | | | |
| Research Income and any Significant Changes (RISC) | | | | |
| Graduate School Establishment and its activities (GSE) | | | | |

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
