# Peer review of "Integrated Reporting Implementation and Core Activities Disclosure in UK Higher Education Institutions"

_admsci, doi:10.3390/admsci11030086_

Round 1

Reviewer 1 Report

The authors prove a very good understanding of the analyzed topic, which is also due to the multiple references used. The authors present their opinions in a concise way.

The article is very well structured, the ideas are presented in a logical order, the conclusions are very well argued.

The article is based on the analysis of statistical data, the choice of variables and regression models  being very well motivated. 

I especially appreciate the interpretation of the results obtained in the context of similar studies that confirm or not the conclusions of the study conducted by the authors. 

I consider that this article stands out through the alarm signal raised regarding the importance of integrated reporting by universities considering the role of leaders that these entities have in the economy ( taking in account the revolutions that marked the education system). In addition, mimetism will  generate improvements in companies behaviour, as universities are the real promoters of sustainable development and CSR.

 Pay attention to abstract

 ”The study is based on the annual reports of 369 UK HEIs 12 over three years (2016-2018)” and lines 572-573 ”the academic years 2015-16, 2016-17 and 572 2017-18”.

Pay attention to figure 1, some missing words....

Reviewer 2 Report

The paper “Integrated Reporting Implementation and Core Activities Disclosure in UK Higher Education Institutions” is very interesting. I provide some suggestions. 

In the first part, you refer to the IR and the importance of IT. In the analysis, you should specify how to evaluate this integrated thinking. In conclusion, you say IT is contributing to the enhancement of the level of disclosure on IR content elements. Please, give more information on how this process happens. 

For your study, IR of 369 UK HEIs 12 over three years (2016-2018). Therefore, your focus is on IR. The first reaction is to unveil that so many universities prepare IR (and for three years!)

Given this observation, I wonder the reason why in the initial part of your paper, you talk about also other voluntary reports (sustainability reports, corporate social responsibility). It can be confusing. you investigate the IR and/or the other reports too. I feel to recommend that you focus only on IR and restructure the parts of the paper in a more homogenous way.

I suggest reviewing the abstract to capture the aim of your paper in a better way. 

Attention to some material errors 

Reviewer 3 Report

The paper deal with an interesting and original topic related to the IR and IT. The structure of the paper is clear and robust. The introduction provides sufficient explanation about the aim of the research. The literarature review is very consistent. Methodology is robust and coherent with the aim of the research. Results and conclusions are well defined. 

I suggest to limit the lenght of the contribution: I think 44 pages are more similar to a small book than a research paper. Literature review could be refocused. At the same time methodology can reduced to the most relevant items. 

Best regards 
